


# Contributions of the troposphere and stratosphere to $CH_4$ model biases

Zhiting Wang[1], Thorsten Warneke[1], Nicholas M. Deutscher[1,2], Justus Notholt[1], Ute Karstens[3], Marielle Saunois[4], Matthias Schneider[5], Ralf Sussmann[6], Harjinder Sembhi[7], David W. T. Griffith[2], Dave F. Pollard[8], Rigel Kivi[9], Christof Petri[1], Voltaire A. Velazco[2], Michel Ramonet[10], Huilin Chen[11,12]

(1) Institute of Environmental Physics, University of Bremen, Germany
(2) Centre for Atmospheric Chemistry, School of Chemistry, University of Wollongong, Wollongong, New South Wales, Australia
(3) Max Planck Institute for Biogeochemistry, Germany
(4) Laboratoire des Sciences du Climat et de l'Environnement, France
(5) Karlsruhe Institute of Technology, IMK-ASF, Karlsruhe, Germany
(6) Karlsruhe Institute of Technology, IMK-IFU, Garmisch-Partenkirchen, Germany
(7) Earth Observation Science, Department of physics and Astronomy, University of Leicester, Leicester, UK
(8) National Institute of Water and Atmospheric Research (NIWA), Wellington, New Zealand
(9) Finnish Meteorological Institute Arctic Research Center, FMI-ARC, Finland
(10) Laboratoire des Sciences du Climat et de l'Environnement, LSCE/IPSL, CEA-CNRS-UVSQ, Université Paris Saclay, 91191, Gif-Sur-Yvette, France
(11) Center for Isotope Research (CIO), University of Groningen, Groningen, The Netherlands
(12) Cooperative Institute for Research in Environmental Sciences (CIRES), University of Colorado, Boulder, CO, USA

*Correspondence to*: Z. Wang (zhiting@iup.physik.uni-bremen.de)

## Abstract

Inverse modeling is a useful tool to retrieve $CH_4$ fluxes; however, evaluation of the applied chemical transport model is an important step before using the inverted emissions. For inversions using column data one concern is how well the model represents stratospheric and tropospheric $CH_4$ respectively when assimilating total column measurements. In this study atmospheric $CH_4$ from three inverse models is compared to FTS (Fourier Transform Spectrometry), satellite and in situ measurements. Using the FTS measurements the model biases are separated into stratospheric and tropospheric contributions. When averaged over all FTS sites the model bias amplitudes (absolute model to FTS differences) are 7.4±5.1 ppb, 6.7±4.8 ppb, and 8.1±5.5 ppb in the troposphere for the models TM3, TM5-4DVAR, LMDz-





PYVAR, respectively, and 4.3±9.9 ppb, 4.7±9.9 ppb, and 6.2±11.2 ppb in the stratosphere. The tropospheric model biases show a latitudinal gradient for all models, however there are no clear latitudinal dependencies for stratospheric model biases visible except with the LMDz-PYVAR model. The latitudinal gradient is not present in a comparison with in situ measurements, which is attributed to

the different longitudinal coverage of FTS and in situ measurements. Similarly, a latitudinal pattern exists in model biases in vertical $CH_4$ gradients in the troposphere, which indicates vertical transports of tropospheric $CH_4$ is not represented correctly in the models.

## 1 Introduction

Atmospheric methane ($CH_4$) is the second most important anthropogenic greenhouse gas. Atmospheric

$CH_4$ concentrations began to rise again in 2007 after a decade of near-zero growth (Rigby et al., 2008). Possible explanations for the stability of $CH_4$ concentrations during 1999-2006 include: an increase in anthropogenic emissions and coincident decrease in wetland emissions (Bousquet et al., 2006); decreased northern hemisphere microbial sources (Kai et al., 2011); and a combination of decreasing-to-stable fossil fuel emissions and stable-to-increasing microbial emissions (Kirschke et al., 2013). Several

possible reasons for the renewed growth of $CH_4$ concentrations after 2006 have been proposed including the increase of wetland emissions during 2007 and 2008 in either the tropics, owing to greater than average precipitation, and/or in the Arctic, owing to high temperatures (Dlugokencky et al., 2009), the anthropogenic contribution in the tropics and mid-latitudes in the northern hemisphere during the period 2007-2010 (Bergamaschi et al., 2013), an incrase of emissions from oil- and gas production and use

during 2007-2014 (Hausmann et al., 2016), and from agriculture (Schaefer et al., 2016).

Prediction of the evolution of $CH_4$ in the atmosphere requires knowledge of the sources and sinks. Inverse modeling is usually used to retrieve fluxes from observations of atmospheric concentrations. The commonly used measurements include surface measurements from global networks, such as the NOAA/ESRL (Earth System Research Laboratory of the National Oceanic and Atmospheric

Administration), and total column data from satellites, such as the SCIAMACHY (Scanning Imaging Absorption Spectrometer for Atmospheric Chartography) or GOSAT (Greenhouse gases Observing SATellite). However, compared to total column data the surface measurements characterize the boundary layer only and $CH_4$ concentrations in the boundary layer are sensitive to boundary layer height that is difficult to be accurately simulated in a global transport model. The total column measurements

are less sensitive to model errors in the vertical distributions of $CH_4$, however, they are also only sensitive to broader-scale signatures. Compared to satellite measurements surface in-situ measurements have poor spatial coverage but are more precise and less subject to biases. Total column measurements



of $CH_4$ include a contribution from the stratosphere where the concentrations are influenced by dynamical processes like meridional transport, tropopause variations, and subsidence associated with the polar vortex, and chemistry. If a transport model does not accurately simulate these processes, the retrieved sources and sinks using total column measurements will not be correct (Locatelli et al., 2015a; 2015b). Especially in the polar region, the tropopause height varies strongly and the dynamical processes are complex. Turner et al. (2015) compared GOSAT $CH_4$ with GEOS-Chem simulations, and found large differences at high-latitudes. They proposed that the model bias in total column $CH_4$ at high-latitudes comes from the stratosphere since the validation with TCCON (Total Carbon Column Observing Network), NOAA surface and aircraft measurements, and HIPPO shows good performances of the model in the troposphere. Ostler et al. (2016) assessed accuracies of models in the stratosphere by replacing modeled stratospheric $CH_4$ with satellite measurements. They found that modeled stratospheric $CH_4$ shows large scatter and the corrected total columns of $CH_4$ show improved or degraded agreements with TCCON measurements depending on the used satellites and models. These results imply that satellite-based stratospheric $CH_4$ is not accurate enough to resolve a possible stratospheric contribution to model biases in total column $CH_4$ as uncovered by TCCON. TCCON-based measurements could fulfill such a role, as presented in Saad et al. (2016) and this study. Using HF as a proxy, Saad et al. (2016) derived tropospheric $CH_4$ products and investigated the impact of stratospheric and tropospheric model biases in GEOS-Chem on inversions. They found an increasing stratospheric mismatch with decreasing tropopause altitudes and a phase lag in modeled tropospheric seasonality. A small bias in the modeled $CH_4$ column could come from counteracting stratospheric and tropospheric model errors. They noted that the tropospheric time lag can produce large errors in posterior wetland emissions in high northern latitudes.

In this study the model biases in the stratosphere and troposphere are assessed with respect to the latitudinal pattern. In order to investigate the accuracy of the models several measurements are used: (i) total, tropospheric and stratospheric column-averaged $CH_4$ mole fractions measured at the TCCON (Wunch et al., 2011; Wang et al., 2014), which are used to separate stratospheric and tropospheric contributions to model bias in total columns, (ii) total column-averaged $CH_4$ mole fraction measured by GOSAT (Parker et al., 2011) and $CH_4$ profiles measured by TES (Tropospheric Emission Spectrometer) (Worden et al., 2012), (iii) surface $CH_4$ measured within the NOAA network (Dlugokencky et al., 1994) and (iv) in situ $CH_4$ profiles from aircraft campaign HIPPO (HIAPER Pole-to-Pole Observations) (Wofsy et al., 2011). In the following, Sect. 2 presents the measurements, models and analysis approach, while Sect. 3 presents the results and discussions. Conclusions are drawn in Sect. 4.



## 2 Measurements and models

We work here with near-infrared spectra of TCCON, from which the tropospheric $CH_4$ is derived using an a posteriori correction method in contrast to the direct profile retrieval (Sepulveda et al., 2014) being applied to mid-infrared spectra. The tropospheric $CH_4$ is derived through removing stratospheric
contributions in total column $CH_4$. The stratospheric contributions are estimated from stratospheric $N_2O$ columns derived from total $N_2O$ columns. A calibration of the method against in-situ measurements shows an agreement within $3.0\pm2.0$ ppb (see Figure 1). Given the total and tropospheric $CH_4$ columns, stratospheric column-averaged $CH_4$ is derived using knowledge of the tropopause pressure. The TCCON sites used in this study are listed in Table 1, the products are all using the GGG2014 version
(Wunch et al., 2015), except for at Ny-Ålesund.

The $CO_2$ proxy retrieval method (Frankenberg et al., 2011) is applied in GOSAT data, which infers dry air columns from the $CO_2$ columns retrieved from the same spectra as used in the $CH_4$ retrieval. The GOSAT total column-averaged dry-air $CH_4$ mole fractions used here are version UoL-OCPRv5.1 and only spectra measured in clear sky conditions are used (Parker et al., 2011). GOSAT has a ground
footprint diameter of about 10.5 km and 4 second exposure duration. The TES instrument measures atmospheric radiances from which atmospheric profiles are inferred using an optimal estimation algorithm subject to a priori constraints. The $CH_4$ retrieval of TES has a DOFS (degree of freedom for signal) about 0.8~2.3, which peaks in the tropics and decrease toward high latitudes. The version F07_10 data are applied and measurements with less than 1.4 DOFS are filtered out.

Vertical gradients of tropospheric $CH_4$ can be qualitatively calculated by using the comparative tropospheric column-averaged $CH_4$ and surface $CH_4$. Only long-term time scales are used here, and variations with scales longer than 1.4 years are extracted from the time series of tropospheric and surface $CH_4$. TCCON and in situ sites are selected to be located close to one another so that both instruments measure similar airmasses. The sites and measurements are listed in Table 3.

The $CH_4$ measurements during HIPPO-1 to 5 are those made by a quantum cascade laser spectrometer (QCLS). Calibrations derived through comparisons with NOAA Programmable Flask Package measurements are applied.

The models used in this study are TM3, TM5-4DVAR, LMDz-PYVAR, whose details are given in Table 3. The first two models used a common emission a priori for their inversion runs. Only in situ
measurements at the surface are assimilated. Detailed information on the inversion methodology is discussed in Bergamaschi et al. (2015). The LMDz-PYVAR uses a different a prior and background





stations as constraints, the BG-SP setup described in Locatelli et al. (2015b).

Details about the global atmospheric tracer model TM3 can be found in Heimann and Körner (2003) and the inversion method of the Jena CarboScope is described in Rödenbeck (2005). TM5-4DVAR is a four-dimensional data assimilation system for inverse modeling of atmospheric methane emission (Meirink et al., 2008). The system is based on the TM5 atmosphere transport model (Krol et al., 2005). LMDz-PYVAR is a framework that combines the inversion system PYVAR (Chevallier et al., 2005; Pison et al., 2009) with the transport model LMDz (Hourdin et al., 2006).

For evaluation of the models, we interpolate the simulations in time, latitudes, longitudes and pressure to match the measurements. For the total and tropospheric column-averaged $CH_4$ the model profile is integrated taking the a priori and averaging kernel into account according to Rodgers and Connor (2003) using Eq. 9 and 14 from Wang et al. (2014). In contrast to FTS and GOSAT the transformation of model $CH_4$ profiles to the counterpart of TES is done in logarithms of a prior and model quantities. The NCEP tropopause is used in all calculations, which could not be as accurate for LMDz as for the other two models because LMDz predicts its own meteorology fields through nudging to reanalysis data.

## 3 Comparison between measurements and models

The $CH_4$ column meridional distribution is sensitive to the latitudinal distribution of $CH_4$ sources and sinks, tropopause altitudes, inter-hemisphere transport in the troposphere, and the residual circulation in the stratosphere. Assessing latitudinal variabilities of biases of a model could reveal how well these processes are represented in the model. Another important concern of this study is to determine which of tropospheric or stratospheric model biases contributes more to the total bias. The model to FTS comparison covers the period 2007-2011 when FTS measurements are available and the comparison to GOSAT is for the period 2009-2011.

The latitudinal behavior of the model bias in total column-averaged $CH_4$ mole fractions is revealed by comparisons to FTS and GOSAT measurements as presented in Figure 2. $CH_4$ is emitted mainly in the northern hemisphere, destroyed mainly in the tropics by OH and has a slow inter-hemisphere transport with a temporal scale of approximately 1 year. $CH_4$ is transported into the stratosphere mostly in the tropics and back to the troposphere in the extratropics by the residual circulation. In the troposphere, $CH_4$ concentrations are higher in the northern hemisphere than in the southern hemisphere with a gradient throughout the tropics. In the stratosphere, $CH_4$ has a more or less symmetrical distribution between the two hemispheres. In Fig. 2 the model biases present a clear latitudinal dependence, similar



to results revealed by other studies (e.g. Turner et al., 2015 and Alexe et al., 2015). The latitudinal dependence is similar between FTS and GOSAT northward of 50°S where FTS measurements are available. The model to measurements difference shows a North-South gradient with positive values at northern high-latitude northward of 50°S for all the models.

With FTS-derived tropospheric and stratospheric column-averaged $CH_4$ (Wang et al., 2014) it is possible to examine how the tropospheric and stratospheric columns contribute to the model bias in the total column-averaged $CH_4$. Figure 3 shows yearly and seasonal median model biases scaled by the fraction of the air column in the troposphere and stratosphere. It is clear that model biases in the troposphere exhibit a North-South gradient with positive values in northern high-latitude during all seasons for all

models. In the stratosphere model biases do not present any clear latitudinal pattern that persists through the whole year, and show significant seasonal variabilities for TM3 and TM5-4DVAR. That is consistent with the fact that stratospheric $CH_4$ distributions cycle between summer and winter hemispheric states. In the case of LMDz-PYVAR there is a permanent pattern in the stratospheric biases that is more negative in the south. This pattern is consistent with the North-South gradient in the total

column biases. Comparing to Fig. 2 one can see that the latitudinal pattern of model biases in total column-averaged $CH_4$ results from both the stratosphere and troposphere for LMDz-PYVAR, but arises from the troposphere for TM3 and TM5. The model biases change signs yearly and seasonally, therefore it is more appropriate to use the amplitudes (absolute model to FTS differences) to evaluate the contributions of the troposphere and stratosphere. The medians of model bias amplitudes over all FTS

sites and years are 7.4±5.1 ppb in the troposphere and 4.3±9.9 ppb in the stratosphere for TM3, 6.7±4.8 ppb and 4.7±9.9 ppb for TM5-4DVAR, and 8.1±5.5 ppb and 6.2±11.2 ppb for LMDz-PYVAR.

Evaluations of the models at the surface using in-situ measurements, which are assimilated into the models, show smaller biases than the tropospheric column-averaged $CH_4$. The amplitudes are mostly below 10 ppb in the northern hemisphere except for a few outliers and below 5 ppb in the southern

hemisphere (not shown). The model biases at the surface do not show any significant latitudinal dependence. It is not clear how the model biases at the surface appear in the regions where no measurements are assimilated. However, it could be true that the overestimation of the tropospheric $CH_4$ meridional gradient is due to model biases in the mid and upper troposphere. That would mean that vertical distributions of $CH_4$ in the troposphere are not represented correctly in the models.

Figure 4 presents a comparison of modeled and measured vertical gradients of tropospheric $CH_4$, as qualitatively represented by the difference between the tropospheric column-averaged $CH_4$ and the surface $CH_4$. The vertical gradient is influenced by surface emissions, transport and OH fields.





Generally there are negative vertical gradients in the northern hemisphere and positive vertical gradients in the southern hemisphere (except for over the southern continents in locations with strong emissions). Here we refer to decreasing $CH_4$ mole fractions with altitude as a negative vertical gradient, while increasing $CH_4$ with altitude is a positive vertical gradient. This occurs because most $CH_4$ is emitted in the northern hemisphere and mixed into the southern hemispheric Hadley cell, whose southward branch prevails in the mid and upper troposphere. In the troposphere, surface emissions cause decreasing $CH_4$ with altitude, while OH oxidation causes a negative vertical gradient. The model biases in the tropospheric vertical gradient are mostly positive in mid and high northern latitudes, and negative at other latitudes. So the overestimated tropospheric $CH_4$ in mid and high northern latitudes could not originate from overestimated emissions, which should result in a more negative vertical gradient in the troposphere.

Figure 5 shows a comparison between model simulations and HIPPO measurements. The results are longitudinally averaged for all five HIPPO missions within grids of 4° latitude and pressure increments of 10 hPa. A significant feature is an overestimation of $CH_4$ in the lowermost stratosphere over latitudes higher than 30°S/N, much larger than the biases in the troposphere. It is not clear whether the overestimation arises from the residual transport in the stratosphere, which appears to be too strong, a too high tropopause, an incorrect vertical $CH_4$ gradient across the tropopause or misrepresentation of stratospheric chemistry. Underestimations dominate in the southern troposphere, especially in the upper southern troposphere, consistent with the results in Fig. 4 that modeled gradients of tropospheric $CH_4$ are biased negative as revealed by FTS and surface measurements. There are no significant patterns for the vertical gradient bias in the northern troposphere.

Unlike for the FTS, the model biases in the tropospheric column-averaged $CH_4$ revealed by HIPPO do not show a significant latitudinal trend (Fig. 6, only TM3 are shown there since other models gives similar behavior). This could be because the FTS measured tropospheric $CH_4$ is defined differently than the mean mole fraction between the surface and thermal tropopause. In deriving the FTS tropospheric $CH_4$, the stratospheric $CH_4$ is removed via its linear correlation with $N_2O$. The tropopause in the FTS data therefore has a chemical definition. It is not clear how different from each other the two kinds of tropopause are during this period. A sensitivity test was conducted by shifting the thermal tropopause 200 hPa upward to include the lower stratosphere where $CH_4$ is overestimated by the models. The model biases compared against HIPPO then become closer to those against FTS. However, this difference of 200 hPa between the chemical and thermal tropopause is unrealistically large. In addition, the FTS measured tropospheric $CH_4$ agrees well with in situ measurements in Fig. 1 where the thermal tropopause is applied.





Another possible explanation is that HIPPO sampled the atmosphere mostly in the region 150°E~110°W, over the Pacific Ocean. Apart from Izaña and Ny-Ålesund, the northern FTS sites are located inland. The longitudinal dependence of model biases is investigated with TES measured $CH_4$ mole fractions at 215, 464 and 680 hPa (the lower panel in Fig. 6). Because the TES profiles have limited vertical resolution, the concentrations at the three levels are not independent. The weighting function of $CH_4$ at 215 hPa peaks around 200 hPa in the tropics and around the 300 hPa higher than 50°N/S. The measurements at 464 hPa show the largest sensitivity around 500~600 hPa, and those at 680 hPa have similar vertical sensitivity but less weights above 400 hPa. The comparisons are separated into a region representing HIPPO sampling (referred as region I) and the remaining longitudes (referred as region II). Differences between the model biases in the two regions occur northward of 45°N most significantly at the level 215 hPa. Increases in the model biases continue in region II but decrease in region I, which is more or less similar to the differences between model biases revealed by FTS and HIPPO in these latitudes. Consistent with FTS the model-TES difference also shows a North-South gradient northward of 50°S. However, it is not clear whether the latitudinal pattern comes from the TES retrieval or model errors. Validation of TES tropospheric $CH_4$ with HIPPO gives near zeros biases except for latitudes 40°~60°N where the TES biases vary in -10~-20 ppb (Herman et al., 2014).

## 4 Conclusions

In this study, three inverse models for $CH_4$ are evaluated using different observations that cover different scales. The aim is to determine whether most of the model biases are located in the stratosphere or troposphere. With FTS stratospheric and tropospheric column-averaged $CH_4$, retrieved from total column FTS measurements, it is shown that model bias amplitudes are 7.4±5.1 ppb, 6.7±4.8 ppb, and 8.1±5.4 ppb in the troposphere for TM3, TM5-4DVAR, and LMDz39-PYVAR. The corresponding stratospheric biases are 4.3±9.9 ppb, 4.7±9.9 ppb, and 6.1±11.2 ppb, respectively. The tropospheric model bias exhibits a North-South gradient northward of 50°S with an overestimation in northern high-latitude for all models. There is no persistent latitudinal pattern with season in the stratospheric model bias for TM3 and TM5-4DVAR.

The evaluation of the models at the surface shows a smaller bias compared to the tropospheric column-averaged $CH_4$. We assume that the tropospheric model biases are mainly located in the middle and upper troposphere although comparisons at the surface are only limited to sites where the measurements have been assimilated into the models. Comparison with HIPPO in the troposphere does not show the same latitudinal pattern in model biases as in the comparison with FTS. Two possible reasons are suggested: (i) the difference between the thermal tropopause and that in the FTS tropospheric $CH_4$




product, (ii) the latitude patterns of model biases are dependent on longitude. Using an assessment of model biases relative to TES satellite measurements, we propose that the longitudinal dependence of the model performance contributes to the difference between HIPPO and FTS. However, the tropopause altitude could cause differences during short temporal scale processes, e.g. stratospheric intrusions where the stratospheric air can sink below the thermal tropopause. Stratospheric air can also detach from the stratosphere completely and enter the troposphere. If the detached air parcels still have stratospheric properties, e.g. $CH_4$ correlates with $N_2O$ as in the stratosphere, the FTS measured tropospheric $CH_4$ would exclude these air parcels; however, direct integration from the surface to the thermal tropopause, such as that used for the models and in situ profiles will include these in the tropospheric $CH_4$. More confusing situations could occur where there is strong mixing across the UTLS (the upper troposphere and lower stratosphere) and both thermal and chemical tropopause are not well defined. Future works will be devoted to clarifying the realistic content in FTS tropospheric $CH_4$ and to defining a reasonable approach to comparing it with in situ and model products in these situations.

## Acknowledgements

This research is funded by EU project InGOS. TCCON data were obtained from the TCCON Data Archive, hosted by the Carbon Dioxide Information Analysis Center (CDIAC) - tccon.onrl.gov. The TM5-4DVAR data is from Peter Bergamaschi at European Commission Joint Research Centre, Institute for Environment and Sustainability, Italy. Lamont-AirCore measurements have been provided by the Colm Sweeney at the NOAA Carbon Cycle and Greenhouse Gas Group Aircraft Program (http://www.esrl.noaa.gov/gmd/ccgg/aircraft/). Nicholas Deutscher is supported by an ARC-DECRA Fellowship, DE140100178. TCCON measurements at Park Falls and Lamont are possible thanks to NASA grants NNX14AI60G, NNX11AG01G, NAG5-12247, and NNG05-GD07G, and the NASA Orbiting Carbon Observatory Program, as well as technical support from the DOE ARM program (Lamont) and Jeff Ayers (Park Falls). Darwin and Wollongong TCCON support is funded by NASA grants NAG5-12247 and NNG05-GD07G and the Australian Research Council grants DP140101552, DP110103118, DP0879468 and LP0562346, as well as support from the GOSAT project and DOE ARM technical support in Darwin. The EU projects InGOS and ICOS-INWIRE and the Senate of Bremen provide financial support for TCCON measurements at Bremen, Orleans, Bialystok and Ny Alesund, and Orleans is also support by the RAMCES team at LSCE. The Lauder TCCON programme is core-funded by NIWA through New Zealand's Ministry of Business, Innovation and Employment.



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

Table 1. Overview of TCCON sites used.

| TCCON site | Latitude/°N | Longitude/°E | Altitude/masl | Citation |
| --- | --- | --- | --- | --- |





| Ny-Ålesund | 78.9 | 11.9 | 20 | Messerschmidt et al., 2010 |
|---|---|---|---|---|
| Sodankylä | 67.3668 | 26.6310 | 188 | |
| Bialystok | 53.23 | 23.025 | 183 | Messerschmidt et al., 2012 |
| Bremen | 53.10 | 8.85 | 27 | Messerschmidt et al., 2010 |
| Orléans | 47.97 | 2.113 | 130 | Messerschmidt et al., 2010 |
| Garmisch | 47.476 | 11.063 | 740 | Sussmann et al., 2013, Sussmann and Rettinger 2014. |
| Park Falls | 45.945 | -90.273 | 440 | Washenfelder et a., 2006 |
| Lamont | 36.604 | -97.486 | 320 | Wunch et al., 2009 |
| Izaña | 28.3 | -16.483 | 2370 | Blumenstock et al., 2014 |
| Darwin | -12.424 | 130.891 | 30 | Deutscher et al., 2010 |
| Wollongong | -34.406 | 150.879 | 30 | Deutscher et al., 2010 |
| Lauder | -45.038 | 169.684 | 370 | Sherlock et al, 2014 |

Table 2. Information on the models and setup details.





| Model | Institute | Resolution (lat×lon) | No. of vertical levels | Output time step (hour) | Meteorology |
|---|---|---|---|---|---|
| TM3 | Max Plank Institute for Biogeochemistry | 4°×5° | 26 | 3.0 | ERA-Interim |
| TM5-4DVAR | European Joint Research Centre | 1°×1° for Europe, 6°×4° for the rest of the world | 25 | 1.5 | ECMWF-IFS |
| LMDz-PYVAR | Laboratoire des Sciences du Climatet de I'Environment | 1.875°×3.75° | 39 | 3.0 | Prediction from LMDz |

Table 3. FTS and in-situ sites used for comparison to FTS tropospheric column-averaged $CH_4$ and surface/tower $CH_4$.

| FTS site | | | | In situ site | | | |
|---|---|---|---|---|---|---|---|
| Name | Lat/°N | Lon/°E | Alt/masl | Name | Lat/°N | Lon/°E | Alt/masl |
| Ny-Ålesund | 78.923 | 11.923 | 24 | zep/NOAA | 78.907 | 11.889 | 479 |
| Sodankylä | 67.367 | 26.631 | 188 | pal/NOAA | 67.970 | 24.120 | 565 |
| Orléans | 47.965 | 2.113 | 132 | Trainou tower | 47.965 | 2.113 | 311 |
| Park Falls | 45.945 | -90.273 | 440 | lef/NOAA | 45.930 | -90.270 | 868 |
| Lamont | 36.604 | -97.486 | 320 | sgp/NOAA | 36.620 | -97.480 | 374 |
| Izaña | 28.300 | -16.483 | 2370 | izo/NOAA | 28.300 | -16.480 | 2378 |
| Lauder | 45.038 | 169.684 | 370 | bhd/NOAA | -41.408 | 174.871 | 90 |





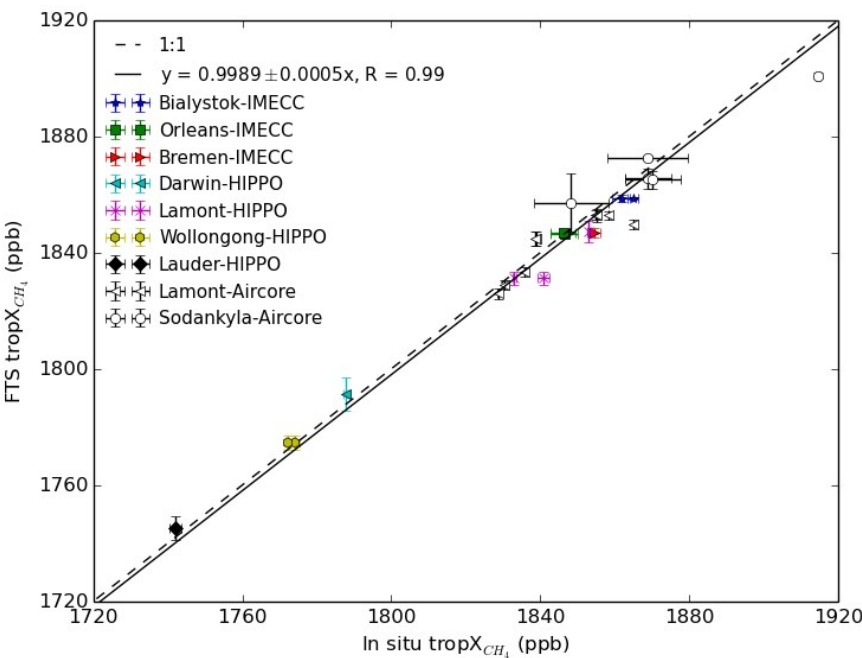

Figure 1. Calibration results of FTS derived tropospheric column-averaged $CH_4$ mole fractions against in situ measurements. The in situ profiles are smoothed using GFIT $CH_4$ averaging kernels in the troposphere as described in Wang et al. (2014). The FTS data are averaged for the in situ measurement periods.





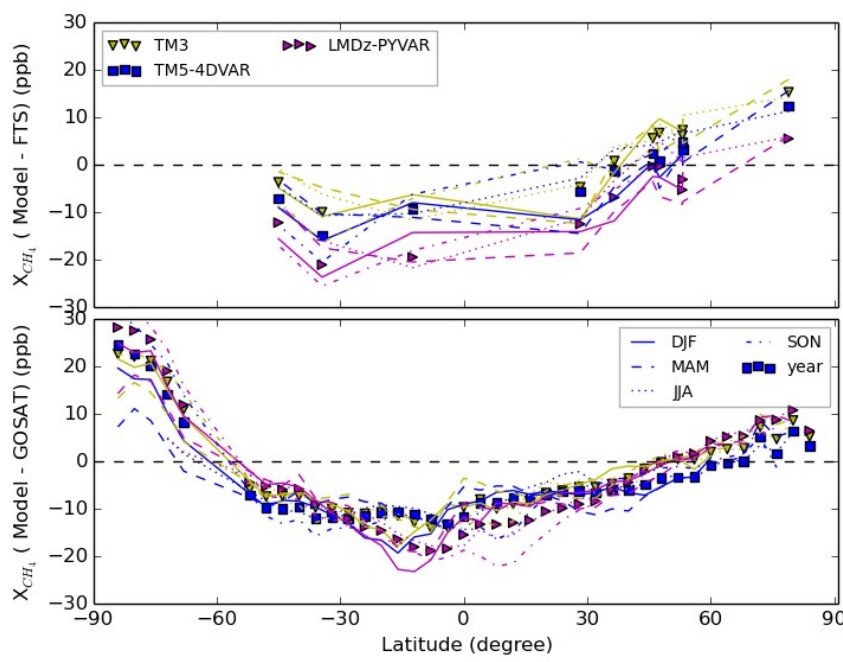

Figure 2. Yearly and seasonal mean model bias of total column-averaged $CH_4$ mole fractions plotted as a function of latitude. The upper panel is the results using FTS data while the lower panels is for GOSAT. The difference for the models is given in yellow (TM3), blue (TM5-4DVAR), and magenta (LMDz-PYVAR). The average of FTS results is for the period 2007-2011 where FTS measurements are available, and for GOSAT in the period 2009-2011.




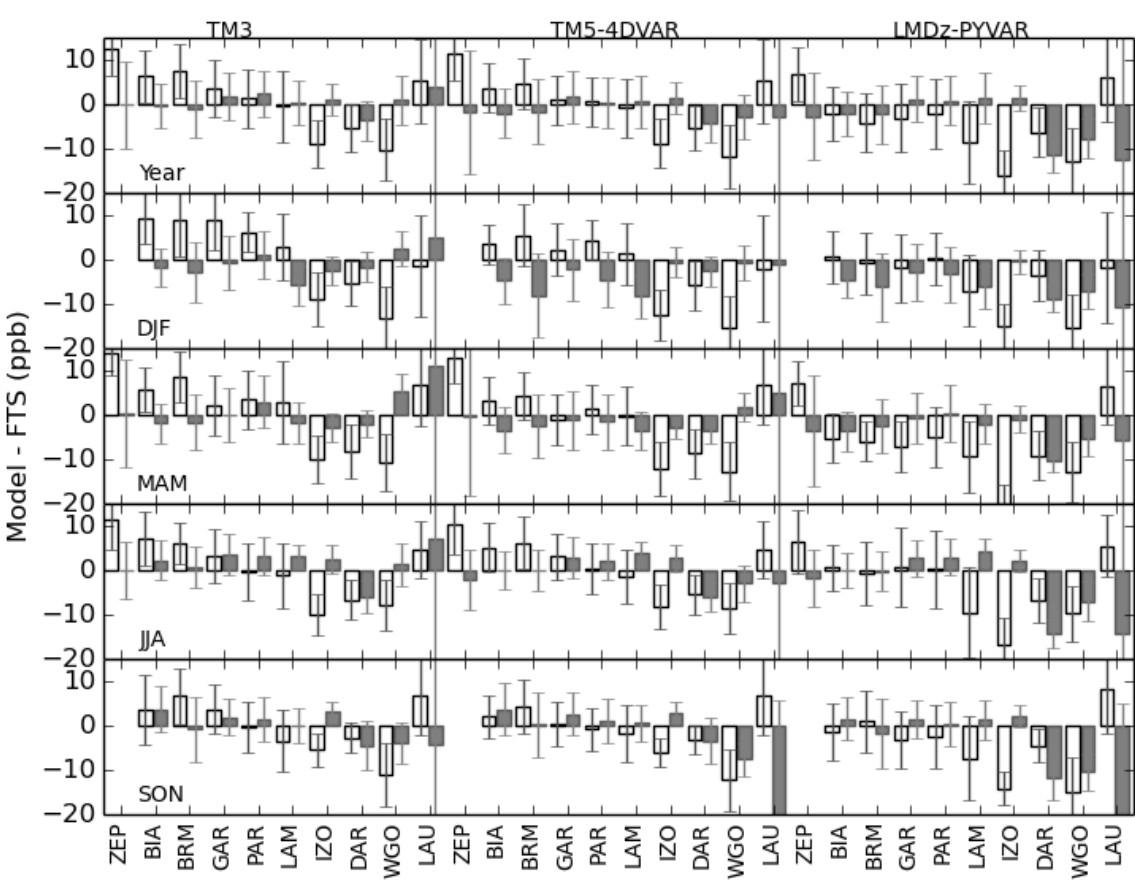

Figure 3. Yearly and seasonal medians of the scaled stratospheric and tropospheric contributions in modeled total column biases at TCCON sites. The sites from left to right is North to South. The white bar denotes the tropospheric bias, the grey bar for the stratospheric bias. The scale factor for the model bias are the air column fractions $P_t/1000$ (stratosphere) and $(1-P_t/1000)$ (troposphere), where $P_t$ is the




tropopause pressure. The error bar are the standard deviations of the model biases. The results are averaged for 2007-2011 when FTS measurements are available.

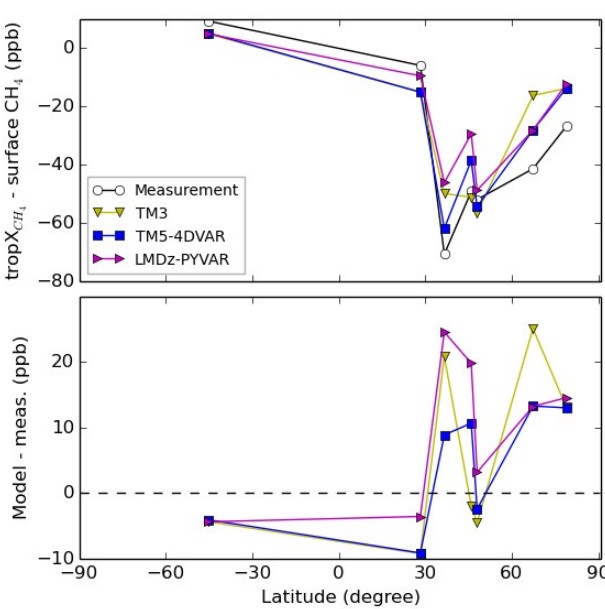

Figure 4. Measured (black) and simulated (yellow: TM3, blue: TM5-4DVAR, LMDz-PYVAR: magenta) vertical gradients of $CH_4$ in the troposphere (top panel) and differences between the measurement and simulations (lower panel) against latitude. The results are averaged for 2007-2011 when FTS 10 measurements are available.





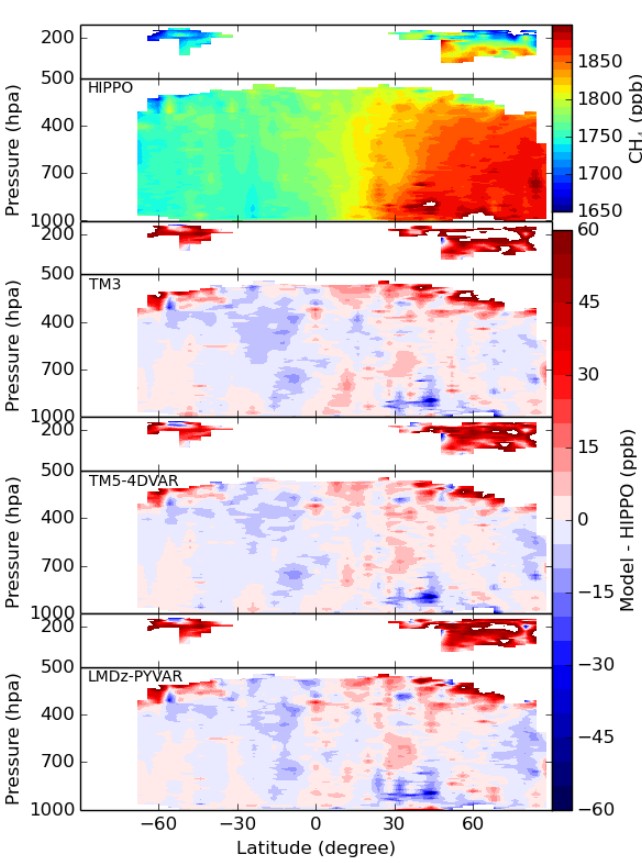

Figure 5. HIPPO measured $CH_4$ and differences with models in the stratosphere (short panel) and
10   troposphere (high panel). The result is an average for five HIPPO missions, averaged for latitudinal bins
of 4° and vertical increments of 10 hPa.




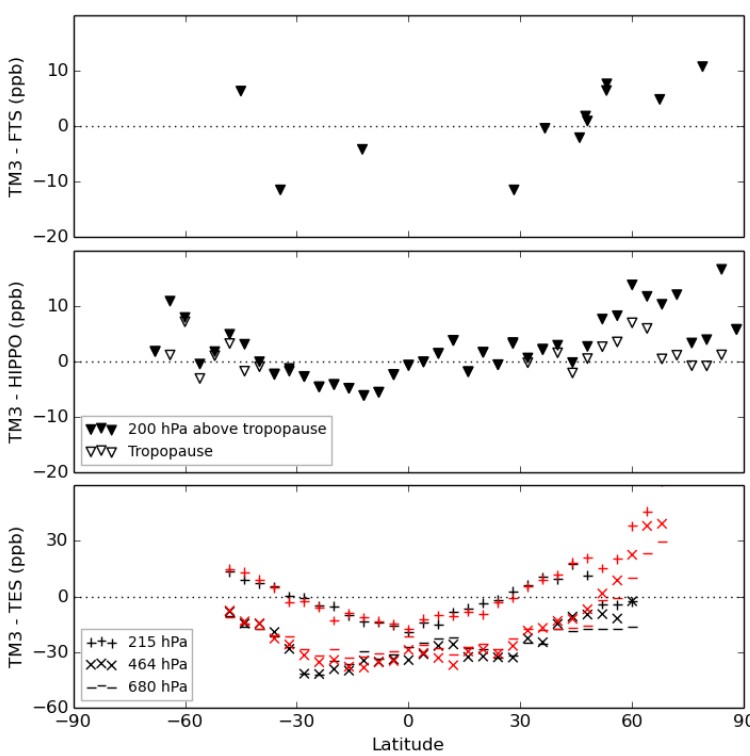

Figure 6. Comparisons of $CH_4$ between TM3 and (upper panel) FTS, (middle panel) HIPPO and (lower
10    panel) TES. In the case of HIPPO and FTS tropospheric column-averaged $CH_4$ is compared, which is
obtained from integration between surface and the tropopause (empty characters) or 200 hPa above the





tropopause shifted (solid characters). For TES $CH_4$ mole fractions at 215 hPa, 464 hPa and 680 hPa are compared with TM3 simulations in a region 110°W~150°E (red) and the region beyond it (black) separately. Both TM3 and measurements are averaged during HIPPO 1-5 period.