# Peer review of "Contributions of the troposphere and stratosphere to CH4 model biases"

_Atmospheric Chemistry and Physics, 2016_

## Referee Comment (RC1) · Anonymous Referee #2 · 20 Jan 2017

This paper evaluated posterior atmospheric CH4 concentrations from three inverse models using in-situ, TCCON and satellite observations. The results show systematic model biases at troposphere and stratosphere, which could adversely affect the assimilation of total column data. Overall the paper is well written, and their results are interesting. It should be accepted for publication after minor revisions.

Major comments: 1. The paper is focused on model evaluations. But no necessary detail on these models such as the surface CH4 fluxes, meteorological fields, model resolution, and chemistry scheme are presented (although references are provided), or are used to explain their different performances (for example Figure 3).

2. Biases in GOSAT retrievals, and their implications on the model evaluations have not been discussed by the authors. GOSAT XCH4 has not been fully validated (particularly)

[Figure]

over tropical regions, and could itself have latitude-dependent bias as well. I suggest the authors use more recent version of GOSAT XCH4 retrievals (such as OCPR v7) as well. Minor comments: 1. Line 37, Page 1: '...6.2+/-11.2' ppb in the stratosphere ...' The notation of +/-11.2 ppb may be mis-leading, as in this case the 'amplitude' as defined by the authors could not be negative.

2. Figure 1: Caption and main text does not provide necessary information, for example, the information about IMECC, and aircore data etc.

3. Line 6, Page 4: '...infers dry air columns from the CO2 columns retrieved from the same spectra as used in the CH4 retrieval'

The sentence is not clear, and no mention of model CO2 concentrations, which is one of the possible sources for biases in GOSAT proxy XCH4 data.

4. Line 14, page 4: '...F07_10 data are applied and measurements with less than 1.4 DOFS are filtered out..', More detailed information such as the observation coverage and errors will be helpful.

5. Line 30, Page 5: '...Figure 3 shows yearly and seasonal median model biases scaled by the fraction of the air column in the troposphere and stratosphere...'.

I suggest adding the number of the TCCON observations at different months to the plot. Also it is interesting to know whether TCCON retrievals have biases depending on the solar zenith angles.

6. Line 5, Page 6: '...one can see that the latitudinal pattern of model biases in total column-averaged CH4 results from both the stratosphere and troposphere for ...' Some explanation of different performances of the three models shown in Figure 3 in terms of surface fluxes, transport or chemistry scheme will be helpful.

7. Line 20, Page 6: TCCON and in situ sites are selected to be located close to one another so that both instruments measure similar airmasses ...'

The TCCON and in-situ measurements have different measurement frequencies. For example, availability of TCCON data usually has strong seasonal variations. How will these differences affect the results presented in Figure 4?

8. Table 3: typo: The latitude of the Lauder TCCON site should be -45.038.
* * *

---

## Referee Comment (RC2) · Anonymous Referee #1 · 1 Feb 2017

This study investigates the origin of a latitudinal bias that has been reported in a number of inverse modelling studies of CH4 using GOSAT retrievals. Most of those studies point to the stratosphere as the plausible origin of a model bias. This study, however, suggests that the upper troposphere could make a sizeable contribution, pointing to errors in the model representation of vertical transport as a possible cause. The comparisons that are presented make a useful contribution to the discussion. However, in my opinion, and as will be explained below, their interpretation requires further attention. This needs to be solved to make this study suitable for publication in ACP.

GENERAL COMMENTS

As explained on page 6, biases are assessed by taking the absolute difference between model and FTS. The motivation is that biases may change sign seasonally, and therefore may not show up in annual averages when positive and negative contributions cancel out. However, whether this is a good choice or not depends on the kind of bias that is investigated. Here the focus is largely on a latitudinal bias. Suppose that there is no latitudinal bias in the annual mean, but only a latitudinally varying bias in the seasonal amplitude. By taking absolute model to FTS differences across the year you would end up with a latitudinally varying bias. In this case the choice of absolute differences was clearly not appropriate. There may not be a single solution to this problem for the biases that are investigated here, but the meaning of the numbers that are summarized in the abstract and the conclusions for stratospheric and tropospheric contribution to the bias is not clear to me. A relation with a latitudinally vaying bias is suggested, but do these numbers really reflect stratospheric and tropospheric contributions to that bias. This requires more attention, including information on how the absolute differences are calculated (on every data point like an RMS, or on monthly averages, or?).

According to the caption of Figure 3, the tropospheric and stratospheric model biases are scaled with the corresponding contributions of the troposphere and the stratosphere to the total column air mass. However, there is a danger in doing so. Suppose that the model had a latitudinally and seasonally uniform offset in the tropospheric concentration. Then the scaling with the seasonal and latitudinal varying tropopause pressure would introduce a seasonal and latitudinal variation in the bias. In that case, when you look for varying biases within the troposphere in comparison with in situ data you wouldn't find any. This is exactly what seems to be happening here. This problem is attributed to differences in the global representation of the measurements, but could also be caused by differences in the NCEP and N2O derived tropopause heights. Since CH4 shown show a sharp vertical concentration gradient just above the tropopause, the analysis may be quite sensitive to how these heights compare. The uncertainty of this needs to be assessed and discussed.

The comparison with TES is used to investigate longitudinal variations in the bias and the global representativeness of the comparisons with HIPPO which are limited to the Pacific. Apart from the fact that it is not clear that the TES data for the troposphere are accurate enough for this purpose (sizeable offsets are seen in the troposphere, that are not due to the TM3 model), the results do not seem to support the case that is made. If anything, the latitudinal gradient in the offset is stronger in the Pacific longitude band (in red) then at other latitudes. The authors are right that the bias has a longitudinal dependence, but it

works on the wrong direction. This needs to be discussed more clearly, and the message of the study should be brought in accordance with this finding.

Looking at Figure 5, the most significant differences between the models and HIPPO seem really at the highest measured altitudes. You might debate whether they are in the troposphere or the stratosphere. I wonder how important this really is. Wouldn't it be better to conclude that the problems show up most strongly at tropopause altitudes. In that case the method of separating the troposphere from the stratosphere may actually not be so appropriate. A plausible cause could be strat-trop exchange. I don't see how the results that are presented here exclude this possibility. Yet, it is not considered as an option.

SPECIFIC COMMENTS

page 4, line 8: Where does the tropopause pressure come from?

page 4, line 13: What model $CO_2$ fields are used to translate the retrieved ratios into $XCH_4$?

page 5, line 13: 'The NCEP tropopause ...'. It is less accurate for TM5 also, which doesn't use NCEP either (in TM3 it depends on the meteo that was used). Please reformulate to make this sentence more accurate.

Page 7, line 18: 'underestimations dominate'. There are lower values elsewhere, so it is not clear that they 'dominate' in the SH.

Figure 3: Please add vertical lines between the columns (i.e. models). At the boundary between the models it is not so clear which bar belongs to which model.

Page 6, line 1: It would be fair to add Monteil et al, JGR, 2013 here, since they were among the first to report a latitudinal bias.

TECHNICAL CORRECTIONS

page 2, line 4: 'transport' i.o. 'transports'

page 2, line 19: 'increase' i.o. 'incrase'

page 4, line 11: '$CH_4$' i.o. '$CO_2$'

page 4, line 11: 'applied to' i.o. 'applied from'

Page 7, line 2: 'except over' i.o. 'except for over'

Figure 4: the dashed zero line is missing in the upper panel

Page 7, line 23: 'show' i.o. 'gives'

---

## Author Comment (AC1) · 31 Mar 2017

Major comments: 1. The paper is focused on model evaluations. But no necessary detail on these models such as the surface CH4 fluxes, meteorological fields, model resolution, and chemistry scheme are presented (although references are provided), or are used to explain their different performances (for example Figure 3).

All the three models optimize CH4 field against in situ measurements at the surface through inversions of the CH4 emissions. The chemical reactions considered in the models are the oxidation by OH in the troposphere, and by Cl, OH and O(1D) in the stratosphere. The fields of the radicals are prescribed monthly with no interannual changes. A description added in the first paragraph in page 5 to illustrate those. The meteorological fields, model resolution are included in Table 2.

[Figure]

2. Biases in GOSAT retrievals, and their implications on the model evaluations have not been discussed by the authors. GOSAT XCH4 has not been fully validated (particularly)over tropical regions, and could itself have latitude-dependent bias as well. I suggest the authors use more recent version of GOSAT XCH4 retrievals (such as OCPR v7) as well.

We follow the advise of the referee, and the version OCPR v7 of GOSAT data is used in the Figure 2.

Minor comments: 1. Line 37, Page 1: '... 6.2+/-11.2' ppb in the stratosphere ... ' The notation of +/-11.2 ppb may be mis-leading, as in this case the 'amplitude' as defined by the authors could not be negative.

The symbol '+/-' has been changed to '$\pm$' in the text.

2. Figure 1: Caption and main text does not provide necessary information, for example, the information about IMECC, and aircore data etc.

The FTS data are averaged for the in situ measurement periods. The IMECC is an aircraft campaign over Europe (Geibel et al., 2012). The Lamont-AirCore measurements are from Greenhouse Gas Group Aircraft Program (http://www.esrl.noaa.gov/gmd/ccgg/aircraft/). The AirCore data at Sodankylä is from the FTS group there. Following this advise, the information has been added to the caption of Fig. 1.

3. Line 6, Page 4: '... infers dry air columns from the CO2 columns retrieved from the same spectra as used in the CH4 retrieval' The sentence is not clear, and no mention of model CO2 concentrations, which is one of the possible sources for biases in GOSAT proxy XCH4 data.

The sentence has been changed to ' .... spectra as used in the CH4 retrieval. This method assumes the CO2 concentrations are known and provided by model simulations. ' at line 16 in page 4.

4. Line 14, page 4: '... F07_10 data are applied and measurements with less than 1.4 DOFS are filtered out...', More detailed information such as the observation coverage and errors will be helpful.

Validation of F07_10 data against to HIPPO measurements shows a bias of -8 ∼ 5 ppb with standard deviations of 25 ∼ 50 ppb below 100 hPa (Herman and Osterman, 2014). According to the advise of the referee the information has been added to the line 24∼25 in page 4.

5. Line 30, Page 5: '... Figure 3 shows yearly and seasonal median model biases scaled by the fraction of the air column in the troposphere and stratosphere ...'. I suggest adding the number of the TCCON observations at different months to the plot. Also it is interesting to know whether TCCON retrievals have biases depending on the solar zenith angles.

There are 10 sites used in that plot, for all seasons, except for measurements at ZEP is absent during the season DJF and SON. This is clearly seen in Fig. 3. The TCCON products has been corrected for solar zenith angle dependence. So this bias should be minor.

6. Line 5, Page 6: '... one can see that the latitudinal pattern of model biases in total column-averaged CH4 results from both the stratosphere and troposphere for ...' Some explanation of different performances of the three models shown in Figure 3 in terms of surface fluxes, transport or chemistry scheme will be helpful.

All the models are optimized with respect to surface measurements already. So the surface emission might not been the main reason to the different performances. The tropospheric oxidation also directly influence the surface CH4 concentrations, and the optimization process should give emissions consistent with the prescribed OH field in each model. The only significant difference among the models could come from convection, the North-South transport, and the transport from the troposphere to the stratosphere. However, it is difficult to give some useful discussion since only column

measurements are used to evaluate the models.

7. Line 20, Page 6: TCCON and in situ sites are selected to be located close to one another so that both instruments measure similar airmasses ::: The TCCON and in-situ measurements have different measurement frequencies. For example, availability of TCCON data usually has strong seasonal variations. How will these differences affect the results presented in Figure 4?

Yes, the TCCON measurement has a different sampling frequencies compared to the in situ measurement. In our analysis the measurement series of TCCON and in situ has been filtered to extract variations with temporal scale longer than 1.4 years and only multi-year (longer than 3 years) averaged results are used. Besides, the models are matched to TCCON and in situ measurements in time, respectively, and undergo the same analysis with the measurements. So the model performance against to the measurements should not be affected.

8. Table 3: typo: The latitude of the Lauder TCCON site should be -45.038.

We follow the advise, and this error has been corrected.

---

## Author Comment (AC2) · 31 Mar 2017

The comments by referee #2 mainly include two parts: 1) The tropopause applied to integrate the model CH4 to obtain tropospheric and stratospheric column-averaged CH4 should be check and the results could depend on the definition of the tropopause. 2) The comparison of the model with TES measurements does not support our conclusions.

1) For the first comment some sensitive tests have been conducted, a different definitions for the tropopause are applied to integrate the model outputs. That include, thermal tropopause according to WMO definition, the dynamical tropospause defined as 1.5, 2.5, 3.5 and 4.0 PUV surface in the extratropics and 380 K potential temperature surface in the tropics. The ECMWF-interim reanalysis data is used to calculate

the tropopause.

The sensitive test is applied to TM3 and LMDz-PYVAR (TM5 has a similar configuration with TM3 and then not been tested). These sensitive tests show (see following plots) that, the tropospheric model bias almost is not affected by the selection of the tropopause, even for the unrealistically low tropopause of 1.5 PVU. The amplitude of the stratospheric mode bias changes between the thermal tropoause and the dynamical tropopause of 2.5∼4.0 PVU. However, there is still not a consistent latitudinal gradient existing during whole year for the stratospheric model biases. So the conclusion that the latitudinal gradient in the model bias of total column-averaged CH4 come from the troposphere is valid. The dynamical tropopause of 1.5 PUV give some latitudinal patten in the stratospheric model biases, but, that tropoause is unrealistically low and frequently reach 170 hPb (below 380 K potential temperature surface) in the tropics.

2) For the second comment, the results from TES actually support the conclusion that the inconsistence between the HIPPO and TCCON comparisons with the models come from the longitudinal dependence of latitudinal gradient in the tropospheric model bias. But there are writing errors in the figure caption of the Figure 6 in the manuscript. In the third panel of Figure 6, the black points correspond to HIPPO sampling area (110°W∼150°E) and the red points to the region beyond it.

Please also note the supplement to this comment:
http://www.atmos-chem-phys-discuss.net/acp-2016-1041/acp-2016-1041-AC2-supplement.pdf

**Supplement:**

[Figure]

Figure 1. Yearly and seasonal medians of the scaled stratospheric and tropospheric contributions in modeled total column biases at TCCON sites. The sites from left to right is North to South. The white bar denotes the tropospheric bias, the grey bar for the stratospheric bias. The scale factor for the model bias are the air column fractions $P_t/1000$ (stratosphere) and $(1-P_t/1000)$ (troposphere), where $P_t$ is the thermal tropopause pressure. The error bar are the standard deviations of the model biases. The results are averaged for 2007-2011 when FTS measurements are available.

[Figure]

Figure 2. Same as Fig. 1 except for 4.0 PUV dynamical tropopause is applied.

[Figure]

5    Figure 3. Same as Fig. 1 except for 3.5 PUV dynamical tropopause is applied.

[Figure]

Figure 4. Same as Fig. 1 except for 2.5 PUV dynamical tropopause is applied.

[Figure]

5    Figure 5. Same as Fig. 1 except for 1.5 PUV dynamical tropopause is applied.

---

## Author Comment (AC3) · 5 May 2017

Additional comments: 1. As explained on page 6, biases are assessed by taking the absolute difference between model and FTS. The motivation is that biases may change sign seasonally, and therefore may not show up in annual averages when positive and negative contributions cancel out. However, whether this is a good choice or not depends on the kind of bias that is investigated. Here the focus is largely on a latitudinal bias. Suppose that there is no latitudinal bias in the annual mean, but only a latitudinally varying bias in the seasonal amplitude. By taking absolute model to FTS differences across the year you would end up with a latitudinally varying bias. In this case the choice of absolute differences was clearly not appropriate. There may not be a single solution to this problem for the biases that are investigated here, but the meaning of the numbers that are summarized in the abstract and the conclusions for stratospheric and

tropospheric contribution to the bias is not clear to me. A relation with a latitudinally vaying bias is suggested, but do these numbers really reflect stratospheric and tropospheric contributions to that bias. This requires more attention, including information on how the absolute differences are calculated (on every data point like an RMS, or on monthly averages, or?).

The absolute difference between the models and FTS is only used to calculate the averaged bias over all years and all sites. The results of the absolute bias are only the numbers appearing in Page 6, line 29~30 and in the abstract. The true bias (model-measurements) is used for all other parts of the paper, including all the plots.

2. Looking at Figure 5, the most significant differences between the models and HIPPO seem really at the highest measured altitudes. You might debate whether they are in the troposphere or the stratosphere. I wonder how important this really is. Wouldn't it be better to conclude that the problems show up most strongly at tropopause altitudes. In that case the method of separating the troposphere from the stratosphere may actually not be so appropriate. A plausible cause could be strat-trop exchange. I don't see how the results that are presented here exclude this possibility. Yet, it is not considered as an option.

Yes, the model bias indeed increase abruptly above the tropopause. However, the approach separating the troposphere from the stratosphere does not influence the latitudinal gradient in the model biases of tropospheric $CH_4$ as show in the Fig. 1-5. Only the stratospheric model biases are sensitive to the separation method and appear large when the tropopause is defined as low as 2.5 PUV. But the stratospheric model biases did not present a consistent latitudinal gradient with the model biases in the total columns of $CH_4$.

3. page 4, line 8: Where does the tropopause pressure come from?

In deriving tropospheric $CH_4$ from FTS measured total columns of $CH_4$ and $N_2O$, the linear correlation existing between $N_2O$ and $CH_4$ in the stratosphere is applied. In the

troposphere the N2O concentration is well known and then stratospheric N2O column is obtained through subtracting its tropospheric contributions from the total columns. Because of the correlation stratospheric CH4 column is estimated from N2O columns and finally the tropospheric CH4 column is known. In the process the tropopaue pressure is not needed.

4. page 4, line 13: What model CO2 fields are used to translate the retrieved ratios into XCH4?

The CO2 field is from the CarbonTracker model.

5. page 5, line 13: 'The NCEP tropopause ...'. It is less accurate for TM5 also, which doesn't use NCEP either (in TM3 it depends on the meteo that was used). Please reformulate to make this sentence more accurate.

We redo the analysis with the thermal tropopause derived from ERA-Interim datasets. Now, the sentence has been changed to 'The thermal tropopause calculated using the reanalysis data ERA-Interim is used in all calculations, which could not be so accurate for the TM5 and LMDz models, especially for LMDz that predicts its own meteorology fields through nudging to reanalysis data.'.

6. Page 7, line 18: 'underestimations dominate'. There are lower values elsewhere, so it is not clear that they 'dominate' in the SH.

The sentence has been changed to 'Underestimations dominate in the upper southern troposphere, consistent with the results in Fig. 4 that modeled gradients of tropospheric CH4 are biased negative as revealed by FTS and surface measurements.'.

7. Figure 3: Please add vertical lines between the columns (i.e. models). At the boundary between the models it is not so clear which bar belongs to which model.

We follow the advise of the referee, necessary modifications have been applied to the Fig. 5.

8. Page 6, line 1: It would be fair to add Monteil et al, JGR, 2013 here, since they were among the first to report a latitudinal bias.

The reference is added to the text on page 6, line2.

Technical corrections: 1. page 2, line 4: 'transport' i.o. 'transports' 2. page 2, line 19: 'increase' i.o. 'incrase' 3. page 4, line 11: 'CH4' i.o. 'CO2' 4. page 4, line 11: 'applied to' i.o. 'applied from' 5. Page 7, line 2: 'except over' i.o. 'except for over' 6. Figure 4: the dashed zero line is missing in the upper panel 7. Page 7, line 23: 'show' i.o. 'gives'

All the corrections has been incorporated into the manuscript except for the 6th. In upper panel of Figure 4 most of the values are smaller than zeros, so it is not necessary to draw a zero line there.

---

## Editor Decision (ED2)

Date : 20 June 2017

Dear authors.

Please reply to the remaining comments one-by-one from the reviewer and provide a revised manuscript accordingly.

Wrt to the 2nd comment :
Ref#2 general point : This point was not addressed.
Just to clarify this point from the reviewer that was not addressed in your reply :
To what extent can you exclude that the latitudinal dependent bias is not caused by the scaling approach you take (which can introduce seasonal and latitudinal variations) ?
Please also motivate the applied scaling approach as used in the manuscript.

Best regards, Ilse Aben

---

## Author Response (AR3)

Response to Ref #2

*To what extent can you exclude that the latitudinal dependent bias is not caused by the scaling approach you take (which can introduce seasonal and latitudinal variations) ?*
*Please also motivate the applied scaling approach as used in the manuscript.*

If I understand correctly you mean the scaling applied for tropospheric model bias in Fig. 3.

The start point of the work is that there exist a latitudinal gradient in the bias of the modelled total columns of CH4, as shown in Fig. 2. The purpose of the work is to determine whether the troposphere or the stratosphere contributes to that. The variations of CH4 total columns include the contribution of tropopause variations. To separate the total column into the tropospheric and stratospheric parts., the airmass possessed by each layer must be taken into account except for the CH4 mixing ratios. Similarly we must take the airmass into account when separating the model bias of the total column into separate layers.

The tropopause altitude influences the total and tropospheric columns of CH4 and then its inaccuracy can contribute the corresponded model biases. However, the sensitivity test using several different definitions for the tropopause (AC2 supplement) reveals that the tropopause is not the reason to the latitudinal gradient in the tropospheric model bias.

---

## Editor Decision (ED3)

Date : 12 July 2017

Dear authors,

I urge you to read the comments from the reviewer and my messages more carefully.

1. I asked you to respond to all remaining comments from the reviewer one-by-one. You only responded to the $2^{nd}$ comment from the reviewer. The reviewer had 4 comments.
2. Wrt. your reply on the $2^{nd}$ comment your reply is still not satisfying. It is clear that scaling is meant to weigh the relative contribution from the troposphere and the stratosphere. But the point is to what extent does this influence/determine the results and thereby the conclusions ? So I can only repeat the question as posed in my previous editorial comment : 'To what extent can you exclude that the latitudinal dependent bias is not caused by the scaling approach you take (which can introduce latitudinal and seasonal variations and thus the observed bias) ?' The simulations you refered to in your reply only look at variations in trop.height. That is –I expect- a much smaller effect.
3. The updated manuscript contains all the track changes from the previous revision. Please accept all changes made before, I only want to see the delta changes to the previous version.

Best regards, Ilse Aben

---

## Author Response (AR4)

**Dear Ilse Aben,**

These are reply to your comments.

1. I asked you to respond to all remaining comments from the reviewer one-by-one. You only responded to the 2nd comment from the reviewer. The reviewer had 4 comments.

We think we have responded to all the comments of Ref2 but in two separate files. For clarity they are modified and organized here.

As explained on page 6, biases are assessed by taking the absolute difference between model and FTS. The motivation is that biases may change sign seasonally, and therefore may not show up in annual averages when positive and negative contributions cancel out. However, whether this is a good choice or not depends on the kind of bias that is investigated. Here the focus is largely on a latitudinal bias. Suppose that there is no latitudinal bias in the annual mean, but only a latitudinally varying bias in the seasonal amplitude. By taking absolute model to FTS differences across the year you would end up with a latitudinally varying bias. In this case the choice of absolute differences was clearly not appropriate. There may not be a single solution to this problem for the biases that are investigated here, but the meaning of the numbers that are summarized in the abstract and the conclusions for stratospheric and tropospheric contribution to the bias is not clear to me. A relation with a latitudinally vaying bias is suggested, but do these numbers really reflect stratospheric and tropospheric contributions to that bias. This requires more attention, including information on how the absolute differences are calculated (on every data point like an RMS, or on monthly averages, or?).

The absolute difference between the models and FTS is only used to calculate the averaged bias over all years and all sites. The results of the absolute bias are only the numbers appearing in Page 6, line  $29 \sim 30$  and in the abstract. The true bias (model - measurements) is used for all other parts of the paper, including all the plots. So there are artificial latitudinal varying bias in the plots. The numbers that are summarized in the abstract and the conclusions are meant to give a impression on amplitudes of the model to the measurement difference in the troposphere and stratosphere. The model to measurements differences are calculated on every data point like an RMS.

According to the caption of Figure 3, the tropospheric and stratospheric model biases are scaled with the corresponding contributions of the troposphere and the stratosphere to the total column air mass. However, there is a danger in doing so. Suppose that the model had a latitudinally and seasonally uniform offset in the tropospheric concentration. Then the scaling with the seasonal and latitudinal varying tropopause pressure would introduce a seasonal and latitudinal variation in the bias. In that case, when you look for varying biases within the troposphere in comparison with in situ data you wouldn't find any. This is exactly what seems to be happening here. This problem is attributed to differences in the global representation of the measurements, but could also be caused by differences in the NCEP and N2O derived tropopause heights. Since CH4 shown show a sharp vertical concentration gradient just above the tropopause, the analysis may be quite sensitive to how these heights compare. The uncertainty of this needs to be assessed and discussed.

The start point of the work is that there exist a latitudinal gradient in the bias of the modelled total columns of CH4, as shown in Fig. 2. The purpose of the work is to determine whether the troposphere or the stratosphere contributes to that. The variations of CH4 total columns include the contribution of tropopause variations. To separate the total column into the tropospheric and stratospheric parts, the airmass possessed by each layer must be taken into account except for the CH4 mixing ratios. Similarly

we must take the airmass into account when separating the model bias of the total column into separate layers. The tropopause altitude influences the total and tropospheric columns of CH4 and then its inaccuracy can contribute to the corresponding model biases. However, the sensitivity test using several different definitions for the tropopause (AC2 supplement) reveals that the tropopause is not the reason to the latitudinal gradient in the tropospheric model bias.

The comparison with TES is used to investigate longitudinal variations in the bias and the global representativeness of the comparisons with HIPPO which are limited to the Pacific. Apart from the fact that it is not clear that the TES data for the troposphere are accurate enough for this purpose (sizeable offsets are seen in the troposphere, that are not due to the TM3 model), the results do not seem to support the case that is made. If anything, the latitudinal gradient in the offset is stronger in the Pacific longitude band (in red) then at other latitudes. The authors are right that the bias has a longitudinal dependence, but it works on the wrong direction. This needs to be discussed more clearly, and the message of the study should be brought in accordance with this finding.

The validation of TES measurements in the troposphere does not show a latitudinal bias (Herman and Osterman, 2014). There are some offset but latitudinal gradient in TM3 bias revealed by TES could be reliable. The results from TES actually support the conclusion that the inconsistence between the HIPPO and TCCON comparisons with the models come from the longitudinal dependence of latitudinal gradient in the tropospheric model bias. But there are writing errors in the figure caption of the Figure 6 in the manuscript. In the third panel of Figure 6, the black points correspond to HIPPO sampling area (110° W $\sim$  150° E) and the red points to the region beyond it.

Looking at Figure 5, the most significant differences between the models and HIPPO seem really at the highest measured altitudes. You might debate whether they are in the troposphere or the stratosphere. I wonder how important this really is. Wouldn't it be better to conclude that the problems show up most strongly at tropopause altitudes. In that case the method of separating the troposphere from the stratosphere may actually not be so appropriate. A plausible cause could be strat-trop exchange. I don't see how the results that are presented here exclude this possibility. Yet, it is not considered as an option.

Yes, the model bias in CH4 mixing ratios indeed increases abruptly around the tropopause. However, the approach separating the troposphere from the stratosphere does not influence the latitudinal gradient in the model biases of tropospheric CH4 columns as show in the Fig. 1-5 (supplement of AC2). Only the model biases of stratospheric CH4 columns are sensitive to the separation method and appear large when the tropopause is defined as low as 1.5 PUV. But the model biases in the stratospheric CH4 columns of CH4 columns do not present a consistent latitudinal gradient with the model biases in the total columns of CH4. The strat-trop exchange may introduce variations of CH4 mixing ratio in the troposphere and stratosphere, but our work aims at separating model biases of total CH4 columns into contributions by the troposphere and stratosphere not finding the reasons leading to such biases.

2. Wrt. your reply on the 2nd comment your reply is still not satisfying. It is clear that scaling is meant to weigh the relative contribution from the troposphere and the stratosphere. But the point is to what extent does this influence/determine the results and thereby the conclusions ? So I can only repeat the question as posed in my previous editorial comment : 'To what extent can you exclude that the latitudinal dependent bias is not caused by the scaling approach you take (which can introduce latitudinal and seasonal variations and thus the observed bias) ?' The simulations you refered to in your reply only look at variations in trop.height. That is –I expect- a much smaller effect.

As have been explained, the tropospheric model biases here means the measured tropospheric CH4 column minus modeled counterparts. That depend on the tropospheric CH4 mixing ratio and the tropopause pressure. The purpose of our work is to determine whether the tropospheric or stratospheric CH4 column contribute to latitudinal bias in the total CH4 column. The tropospheric CH4 columns is represented by tropospheric column-averaged CH4 mixing ratios multiplied by (1-Pt/1000) (Pt is the tropopause pressure we divide the columns by 1000 hPa just for easy presentation and does not influence results). Without the scaling the model bias will be evaluated in term of stratospheric and tropospheric column-averaged CH4 mixing ratio. But these quantities are not directly related to the total CH4 column and can not answer question proposed by the result in Figure 2 of the manuscript.

Assuming the surface emission of CH4 is prescribed the total number of CH4 molecular residing in the troposphere will be more or less a constant (fast tropospheric convection compared to slow trop-to-strat transport). However, higher tropopause results in smaller CH4 mixing ratios and lower tropopause for larger mixing ratios. Here we want to evaluate model performances in simulating total number of CH4 molecular in the troposphere. It is not correct to exclude the effects of tropopause pressure or the scaling factor.

So the question 'To what extent can you exclude that the latitudinal dependent bias is not caused by the scaling approach you take (which can introduce latitudinal and seasonal variations and thus the observed bias) ?' is not related to the purpose of our work. The relative contribution of tropospheric and stratospheric airmass is determined by the tropopause pressure. In our simulations, we looked at variations in the tropopause height only because it is the only factor determining the scaling factor regardless of its large or small effects.

3. The updated manuscript contains all the track changes from the previous revision. Please accept all changes made before, I only want to see the delta changes to the previous version.

There are not further modifications relative to the last version yet.

---

## Editor Decision (ED4)

Date : 21 July 2017

Dear authors,

Please realise I am going to give you only one more last chance to adequately deal with the comment from the reviewer. So I urge you to take your time and carefully regard the comment, which I –and the reviewer- consider of serious nature as is reflected in the report scores.

If you need more time to deal with the comment in an appropriate manner please ask ACP to provide more time. I will allow for it.

Best regards, Ilse Aben

---

## Author Response (AR5)

Firstly, we apologize for any misunderstanding in our response to the previous reviews. In this revision, we have tried to make sure that we have clarified any uncertainties in the manuscript and responses. We thank the referees and editor for their patience, as well as their comments.

*Following the correspondence about my review I have analyzed the whole discussion to find the origin if our misunderstanding regarding General point 2 of my review, which had to do with the impact of the scaling of the tropospheric and stratospheric sub columns that is applied.*

*In my opinion the discussion is confused because the authors do not distinguish between the following:*
*1) The contribution of the modelled tropospheric sub column to a bias in the total column averaged mixing ratio*
*2) A bias in the modelled tropospheric sub column averaged mixing ratio*

*If scaling is applied [as mentioned in the caption of figure 3] this is appropriate for 1) but not for 2). However, in the language that is used in the paper there is no distinction between 1) and 2). In many instances the language suggests 2) [e.g. in the abstract: 'the tropospheric model biases show a latitudinal gradient for all models'], referring to numbers that are derived using the scaling method, which doesn't apply to that case and therefore causes confusion.*

Following these comments we have modified the text (highlighted in red color) to clarify if we refer to 1) or 2). We refer to the tropospheric partial column of $CH_4$ only in Figure 3 where we want to evaluate the contribution of the modelled tropospheric partial column to a bias in the total column-averaged mole fraction. At all other places the tropospheric column-averaged mole fraction is used, e.g. the vertical gradient in Figure 4 and the FTS and HIPPO case in Figure 6.

*Special care should be taken when comparing results of 1) to comparisons of the model with in situ data. In situ data can only represent local CH4 mixing ratios. If they are compared to 2) this is problematic already because of the vertical gradient in the troposphere as pointed out by the coauthors. However, they cannot be compared to 1).*

We are aware that the results of 2) can not be directly compared to the results of 1). In Figure 4 we qualitatively represent the vertical gradient of $CH_4$ in the troposphere as the difference between the tropospheric column-averaged mole fraction and surface mole fractions. The vertical gradient is shown to reveal the possible reason for the latitudinal dependence in the model biases of the tropospheric column-averaged mole fraction. In the previous discussion there is much concern on how the scaling approach influences the latitudinal gradients. The following Figure shows the latitudinal distribution of the model biases in both the tropospheric partial column and column-averaged mole fraction.

[Figure]

Figure R1. Latitudinal dependences of yearly averaged model biases in the tropospheric sub column (upper panel) and the tropospheric column-averaged mole fraction (lower panel). The three models are represented by Yellow (TM3), Blue (TM5-4DVAR) and Purple (LMDz-PYVAR) respectively.

*In my original review I tried to explain this point using an example of how the mixing up of 1) and 2) could go wrong: "According to the caption of Figure 3, the tropospheric and stratospheric model biases are scaled with the corresponding contributions of the troposphere and the stratosphere to the total column air mass. However, there is a danger in doing so. Suppose that the model had a latitudinally and seasonally uniform offset in the tropospheric concentration. Then the scaling with the seasonal and latitudinal varying tropopause pressure would introduce a seasonal and latitudinal variation in the bias >>here I meant in XCH4<<. In that case, when you look for varying biases within the troposphere in comparison with in situ data you wouldn't find any. This is exactly what seems to be happening here."*

Previously we did indeed misunderstand the meaning of this comment. It is possible that the tropopause variation could cause latitudinal dependence in the model biases of the XCH4. However from the results in this study the latitudinal dependence of the model biases does not mainly come from the tropopause variation (see Figure R1).

*A good way (and even necessary I would say) way to analyze whether or not the troposphere has a latitudinal/seasonal bias would be to assess the bias between the model and the tropospheric columns derived from TCCON (e.g. in a way similar to what is done in Figure 3, but for tropospheric columns and therefore without scaling).*

The suggestion is exactly what we have tried to implement in Figure 3. The scale factors are used to convert from the FTS measured tropospheric column-averaged mole fractions to the tropospheric partial column.

---

## Author Response (AR6)

Dear Iise Aben,

We have revised the manuscript according to your comments as following.

*Please consider to add the figure in your latest response to the manuscript.*

AC: The figure has been included in the revised manuscript and main text is modified accordingly and highlighted in Red.

Regards
Zhiting and Coauthors